# Importance of Basic Research on the Causes of Feather Pecking in Relation to Welfare

**DOI:** 10.3390/ani10020213

**Published:** 2020-01-28

**Authors:** Lisa B. Fijn, F. Josef van der Staay, Vivian C. Goerlich-Jansson, Saskia S. Arndt

**Affiliations:** 1Division Animals in Science and Society, Animal Behaviour, Faculty of Veterinary Medicine, University Utrecht, 3508 TD Utrecht, The Netherlands; v.c.goerlich-jansson@uu.nl (V.C.G.-J.); s.s.arndt@uu.nl (S.S.A.); 2Division of Farm Animal Health, Faculty of Veterinary Medicine, Behavior & Welfare Group (Formerly Emotion & Cognition Group), University Utrecht, 3584 CL Utrecht, The Netherlands; f.j.vanderstaay@uu.nl; 3Brain Center Rudolf Magnus, Universiteitsweg 100, 3584 CG Utrecht, The Netherlands

**Keywords:** feather pecking, animal welfare, industrial farming, basic research, applied research

## Abstract

**Simple Summary:**

Feather pecking is a problematic behavior shown by commercially held laying hens who peck at and pull out feathers of their pen mates. In effort to battle the economic and welfare issues that result from feather pecking many studies on the subject focus on practical solutions (i.e., they follow an applied research approach), while basic research related to feather pecking, research without a practical aim in mind, has received far less attention. Both applied and basic research contribute to a better understanding of feather pecking behavior. In this article we argue that basic research could make an important contribution to science-based knowledge of the causes of feather pecking and the welfare of hens performing this behavior and of the hens receiving the pecks.

**Abstract:**

Feather pecking is a prominent issue in the commercial egg industry, associated with economic losses and welfare problems. A non-systematic literature search suggests that studies on feather pecking are predominantly concerned with applied research goals. That is to say, they aim to solve or diminish the effects of this problematic behavior by orienting towards practical approaches. The strong emphasis on this research approach has skewed our knowledge of the causes of feather pecking in relation to welfare. While the need for such research is high, there is an equivalent need for basic research that has not received corresponding effort. Also, current research predominantly focuses on the negative effects on the birds being pecked, whereas too little attention is given to the possible welfare problems of the peckers. We argue that more basic research is needed for obtaining comprehensive science-based knowledge of behavioral needs and abilities of hens, in particular with respect to behavioral problems that threaten their welfare.

## 1. Introduction

Industrial animal farming entails management procedures and housing conditions that may compromise animal welfare [1]. The general public as well as farmers, veterinarians, and animal welfare scientists are aware of these problems and invest in their putative solutions, either applying practical approaches or science-based solutions. Whereas this is true for all areas of industrial farming, we will exemplarily address the state of science-based knowledge on feather pecking, a pressing problem in the laying hen industry, as an example.

Almost 363 million laying hens are kept on commercial farms in the European Union [2]. Problematic behaviors such as feather pecking and cannibalism are widespread among these birds. In an effort to battle the economic and welfare issues following from such behaviors, many studies have been carried out. The majority of these studies aim at, and contribute to, solving or diminishing this problem in practice (i.e., these studies comprise applied research questions). Much less research effort has been invested into basic (fundamental) research on the topic, such as questions regarding the underlying mechanisms and brain structures involved in feather pecking, the laying hen’s normal behavioral repertoire, behavioral needs, abilities, affective states, and positive welfare indices. In general, what makes up the (in)ability of the laying hen to adapt to its living conditions? While these questions remain unanswered, we are left with a gap in our scientific knowledge base on why chickens show this problematic behavior.

Feather pecking is a continuing problem in commercial laying hen farming. Severe feather pecking, which consists of forcefully pecking and pulling out feathers of conspecifics, causes infringing welfare issues. This behavior results in feather loss in the recipient of feather pecking and induces tissue injury and pain. The resulting bald patches can entice tissue pecking, a form of cannibalism that may eventually lead to death [3,4]. Moreover, thermoregulation is compromised in the victims, which may lead to chronic discomfort. Feather pecking occurs in all layer housing systems, but the problem is most difficult to control when the birds are housed in larger groups [5]. Social learning facilitates feather pecking and cannibalistic behaviors to transmit through a flock [3,6]. Stocking density and group size appear to have an interactive effect on feather pecking, but the exact mechanisms and causes of such interactions are not yet determined [7].

Following European Union Council Directive 1999/74/EU, a ban on small cage housing systems for laying hens came into effect in 2012 and mega flocks of over a thousand birds have become the standard. A widely practiced ‘solution’ to reduce the impact of feather pecking on the birds receiving the pecks is beak trimming. Beak trimming does not decrease feather pecking per se, but flocks where the hens are beak trimmed show less plumage damage [8,9] through reducing the damage caused by the pecks. Beak trimming may be performed preventively in a flock by trimming the beaks of all the chicks at a very young age. The beak in young birds shows rapid regeneration and in some cases a follow-up trim is performed in older birds [10]. In other cases, it is opted to wait until the birds are older and only trim the beaks of peckers once a feather pecking problem is identified [11]. Beak trimming compromises welfare age-dependently. In very young chicks, during the first days of life, beak trimming is associated with acute pain that appears to subside within a week [12,13]. In older birds there is evidence that beak trimming by method of a hot-blade causes prolonged chronic pain [12]. Additionally, it was found that nerve damage following hot-blade beak trimming can result in reduced sensory abilities and loss of normal beak function, negatively affecting feed and water intake, body weight, preening, and general exploratory behavior [9,14].

In recent years there has been a shift in the technique used to trim beaks. Compared to the traditional hot-blade method, the now widely-used infrared treatment is associated with fewer behavioral changes and less painful tissue damage [12,13,15,16]. Despite the improvement of beak trimming techniques, the remaining welfare concerns have moved several European countries to implement a ban on beak trimming, while other countries have expressed an unwillingness to do so until better alternatives to prevent or control the effects of severe feather pecking are uncovered [8,14].

Numerous studies on the subject of feather pecking behavior and cannibalism in laying hens have been carried out over the years. Following the research & development (R&D) definitions formulated in the Frascati Manual [17], these studies inherently categorize as applied research since they aim to solve or diminish the problematic effects of a species-specific problem in the specific setting of commercial egg production:

“***Applied research** is original investigation undertaken in order to acquire new knowledge. It is, however, directed primarily towards a specific, practical aim or objective*.”[17] (p. 51)

The complements of applied research are pure basic research and basic research:

“***Basic research** is experimental or theoretical work undertaken primarily to acquire new knowledge of the underlying foundations of phenomena and observable facts, without any particular application or use in view*.”[17] (p. 50)

“***Pure basic research** is carried out for the advancement of knowledge, without seeking economic or social benefits or making an active effort to apply the results to practical problems or to transfer the results to sectors responsible for their application*.”[17] (p. 50)

Oriented basic research is considered as an interconnecting link between applied research and the two forms of basic research:

“***Oriented basic research** is carried out with the expectation that it will produce a broad base of knowledge likely to form the basis of the solution to recognized or expected current or future problems or possibilities*.”[17] (p. 51)

An alternative and useful classification was provided by Stokes in what he called Pasteur’s quadrant [18]. Stokes based the classification of research on two questions in a 2 × 2 matrix (Figure 1A). The “use-inspired basic research” in Stokes’ classification is considered as equivalent to the “oriented based research” according to the definition in the Frascati Manual [17]. Use-inspired basic research may bridge the gap between basic and applied research (Figure 1B).

A non-systematic survey of the current literature on feather pecking reveals that, while some studies include oriented basic research questions, the majority of studies pursue applied research goals (Table 1). This skew suggests that the focus on a practical solution may have resulted in fundamental questions regarding feather pecking to remain unanswered and we consequently find ourselves with incomplete information on the causes of feather pecking. Despite extensive research efforts over the past few decades, the causes of feather pecking remain largely unknown [19]. Additionally, the focus understandably has mainly been on the welfare of the victims of feather pecking and solutions to improve their situation. However, a repetitive behavioral disorder such as feather pecking may be indication of negative welfare experienced by the pecking bird that deserves equal attention [20].

There are several studies on the behavioral needs and welfare of laying hens. Commonly, such studies are performed in (small-scale) laboratory or (large-scale) commercial farm settings. To establish the hens’ priorities, their reactions to different elements of commercial farming practices are measured, such as feeding variations, different substrates, floor types, cage/pen sizes, and layouts. Studies investigate the effects on the birds when certain elements are present, absent, or restrictive [21,22]. The results provide insights into activities and environmental factors important to laying hens, ranging from feeding preferences, foraging needs, pre-laying and nesting behaviors, perching needs, adequate space for body movements, and comfort behaviors including preening and dust bathing [21,23]. The findings point to the complexity of the chicken’s preferences, for instance in the case of preferred space where priorities are confounded by interactions between group size and stocking density [23]. The expression of patterns from the hen’s behavioral repertoire depends on genetics, epigenetic effects, and previous experiences and conditions [22]. Negative welfare indices include disease, mortality, decreased egg production, and performance of apparently nonfunctional (maladaptive) behaviors such as pacing, head shaking, feather pecking, spot-pecking, feather sucking, wattle chewing, smothering (many chickens crowding in a small space, sometimes standing on each other, eventually leading to death by suffocation), and high levels of aggression [21,22,23,24].

Welfare studies on the laying hen provide a sound base of scientific knowledge on behavioral priorities of laying hens in commercial conditions and contribute to practical implications that may improve hen welfare. While they answer basic questions on the laying hen’s normal behavioral repertoire and psychological needs, many details remain undiscovered. More knowledge is needed pertaining to the chicken’s abilities (i.e., to the range of situations to which the animal is able to adapt to such an extent that it experiences positive welfare). Also, very little is known about chicken’s affective states and positive welfare indices. Variations between strains need to be considered too. 

With respect to the negative economic impact of feather pecking and the severity of the accompanying welfare concerns, it is imperative to act quickly and look for practical solutions to diminish the effects of this problem. Applied studies on this topic are thus extremely valuable, especially if their results translate to improvements within the scope of industrial farming. Concurrently, basic research is needed to study the causation of feather pecking in relation to behavior and welfare. We expect that a deeper understanding of the causes and mechanisms of feather pecking leads to new angles of vision, new scientific hypotheses, and new approaches to address and prevent this problem behavior.

## 2. Materials and Methods

A non-systematic literature search was performed with the goal of making an inventory of topics related to feather pecking that is being addressed in both applied and basic research. An overview of topics and exemplary papers that address these topics is listed in Table 1, without the pretension that they provide an all-embracing list. These articles are classified according to the type of research categorizations by the Frascati Manual [17] and Stokes’ [18] classification (i.e., they are categorized as pertaining pure basic, oriented basic, and/or applied research goals). Both the Frascati Manual [17] and Stokes’ classification [18] emphasize that basic research by definition is research that is performed without any particular application or use in view. Their classifications also state that basic and applied research have a dynamic relationship (see Figure 1B). A clear categorization of research at times may be difficult. The research type may be identified by the length of time between the study and the application of its results alongside the breadth of application of those results (basic research having a broader application) [17].

## 3. Results

The studies considered in our non-systematic literature search are summarized in Table 1. Additional information about these studies is found in Appendix A. The studies selected provide an exemplary overview of the topics addressed in research about feather pecking, be it putative causes of this behavior, or measures to reduce feather pecking. These publications are listed and summarized under the headers “Environment”, “Genotype”, “Phenotype”, “Physiology”, and “Behavior”. Note that this division is somewhat arbitrary and that a number of studies address topics that can be subsumed under more that one of these headers.

**Environment**: Environmental conditions [11,25] that might affect the development and severity of feather pecking are, among others, stress [26,27], lighting conditions [28,29], and stocking densities [30,31]. Improving environmental conditions such as rearing chicks in dark brooders [32], providing environmental enrichment [8,31,33,34], adapting the diet [29,35] (e.g., dietary tryptophan supplementation [36]), and beak trimming [9] might provide measures to control feather pecking.

**Genotype**: A number of studies addressed strain differences (genotypic differences) regarding feather pecking and feather pecking-related behavior [27,37,38,39,40,41] and identified strains that have a higher propensity to develop feather pecking [29], the age of occurrence of feather pecking, and effects of selection on this behavior [35].

**Phenotype**: A few studies phenotypically characterized feather pecking, using physiological measures and behaviors [42,43].

**Physiology**: The physiology underlying feather pecking (e.g., the effects of serotonin and of dietary manipulation of the serotonin precursor tryptophan [44] of immunological [45] and neurobiological factors [46]) has been assessed.

**Behavior**: Finally, a number of studies related behavior as feather pecking development in selection lines [47], the possible role of social learning [3], and preferences of feather peckers for different substrates (among them feathers) [48].

## 4. Discussion

To assess whether basic research has contributed to our understanding of feather pecking behavior in poultry, we performed a non-systematic survey of literature with the aim of identifying research questions (Figure 1) that were addressed with respect to this topic. The resulting overview (Table 1) supports the notion that only a small proportion of feather pecking related research can be categorized as oriented basic [17] or use-inspired basic research [18], whereas the largest proportion best fits the definition of applied research [17,18]. None of the research unambiguously qualifies as pure basic research [17]. It can be argued that research performed within the framework of an identified practical problem by definition cannot be categorized as pure basic research [17]. Instead, it might be categorized as oriented basic [17], or use-inspired [18], if increased understanding, and not solving practical problems, is the main focus of the research project.

Recently, funding of basic research has tended to decrease in, for example, medical sciences [52], and agricultural sciences, including research with farm animal species as model organisms [53,54]. Areas of research that focus on abnormal functions and behaviors, their underlying causes, and mechanisms, are affected by this funding policy. Calvert stated, “The history of the funding of basic research from the end of the war to the present shows a move away from the belief that scientists should be supported as autonomous truth seekers, whose work will inevitably be beneficial for society, toward the view that scientists should gear their work more directly toward social and economic objectives.” [55], p. 203. Research contributing to a superficial solution of the problem might be favored by funding bodies. Research proposals that aim to elucidate the underlying mechanisms of a problem are less likely receive funding. 

Use-inspired basic research ([18]; see also Figure 1B) is expected to generate knowledge on the causes and mechanisms of problematic behavior such as feather pecking and to direct applied research towards a science-based solution. While the need for applied research on the subject remains high, basic research should receive corresponding effort and funding as to obtain science-based knowledge on the causes of feather pecking and their relation to welfare. 

To properly assess an animal’s welfare state, we need thorough knowledge of the animal’s normal behavioral repertoire and its behavioral needs and abilities [1]. Obtaining that knowledge enables mapping of the underlying motivation for seemingly maladaptive behavior that can be taken as sign of compromised or severely compromised welfare [56]. Measures helping to diminish the expression of feather pecking are desperately needed (for instance, food supplements that affect a reduction in feather pecking [36]). Note, however, that absence of the problematic behavior does not necessarily show that the underlying motivation is no longer present. Considering that the pecking behavior may be indicative of negative welfare, it is important to realize that diminishing occurrence of feather pecking may still leave the pecking birds in a state of negative welfare if the enticing source influencing the underlying motivation is not addressed.

Additionally, it has been suggested that different types of feather pecking (gentle, severe, aggressive) and cannibalistic behaviors are based in different motivational systems [4]. More insight into these motivational systems, their physiological and neurological basis, and their possible interactions is needed to establish sustainable solutions for the welfare problems associated with feather pecking. Without such understanding we risk unknowingly undermining the function of one system by dampening another. For instance, if food supplements or genetic engineering diminish the expression of severe feather pecking, a complete welfare assessment includes determining if this influences the bird’s aggressive pecking behavior and thereby it’s establishment and maintenance of a social hierarchy.

In addition to science-based knowledge of laying hen behavioral needs, affective states, and motivational systems underlying behavioral expressions, we need to learn to detect and recognize the animal’s positive welfare using validated indices [57]. These research questions need addressing if we are to close the gap in our knowledge of feather pecking. Studying the proposed aspects by means of basic research offers more freedom in directions of inquiry. Such freedom is needed at this moment to expand our knowledge of laying hens. It is needed to enable taking a step back from the limiting framework of industrial farming and pursue questions about both the ancestors and current strains of layer hens that may be unrelated to commercial conditions. Similarly, there needs to be room for basic research questions that involve factors from the commercial setting but are formulated in such a way that they pursue pure knowledge of laying hens and causes of feather pecking, without pertaining to orientation towards possible future practical applications. Lending welfare scientists the freedom to formulate research goals outside the scope of commercial farming may lead to fresh ideas and new perspectives with the potential of broad applications currently unknown to us. Fundamental knowledge on the causes of feather pecking in relation to welfare is furthermore needed to facilitate well-informed decisions around the commercial production and consumption of poultry products. 

Feather pecking concerns a number of stakeholders. It has economic implications for farmers because it is associated with decreased egg production, heat loss, higher feed costs, and increased mortality [7]. Supermarkets are involved as they profit from cheap egg production. Product pricing directly affects consumers’ choice. However, the number of consumers questioning the welfare implications of the sold products during production is steadily increasing. Other important stakeholders are European Union (EU) governments when export of animal food products contribute largely to the gross domestic product (GDP) of the state. In the Netherlands for example, agriculture contributes nearly 2% to the national GDP [58] of which over 3% is made up by commercial egg production [59]. Applied research on feather pecking may be the preferred route for stakeholders who are confronted with the economic losses that ensue the issue. Practical applications may alleviate costs in a shorter term than knowledge from basic research will. 

In recent years the public has expressed increasing concern with the welfare of animals kept for food production [60]. Science has an obligation to provide the public with factual knowledge about the welfare state of commercially held hens on a sound scientific basis. The quality of these facts can only be ensured if they result from both applied and basic research, in a balanced matter. Qualitative, well-rounded science-based knowledge would furthermore significantly contribute to the ethical consideration of commercial poultry farming. It is conceivable that increased understanding of the underlying mechanisms of feather pecking will lead us to conclude that the problem will always persist if laying hens are kept under current commercial conditions. We therefore need to consider whether future agricultural practices need to be better adapted to the needs of the respective animal breed rather than vice versa. 

## 5. Conclusions

With this non-systematic review of the literature concerning (severe) feather pecking in chickens, we emphasize the need for basic research regarding the underlying mechanisms involved. So far, the majority of studies have applied research goals, while only few are categorized as oriented basic [17] or use-inspired basic research [18]. Despite the large body of research, the causes of feather pecking remain largely unknown [19]. Questions concerning the adaptive capacity of chickens, their behavioral repertoire and needs, and indices of negative and positive welfare states, would greatly benefit from basic research approaches. Neuronal and physiological mechanisms related to abnormal behavior need to be elucidated. Factors such as gene by environment interactions, resulting in epigenetic modifications, need to be addressed in more detail. In addition, future research should focus on both the victim as well as the pecker, as both might experience compromised welfare. Further insight into the motivational systems underlying the multifactorial problem of abnormal behavior is needed in order to form a basis for prevention and intervention. We exemplarily discussed this topic with respect to feather pecking. However, we argue that more basic research would also, in the long run, benefit the solutions of other welfare problems in commercial farming, for example those recently summarized by Nordquist et al. [1].

## Figures and Tables

**Figure 1 animals-10-00213-f001:**
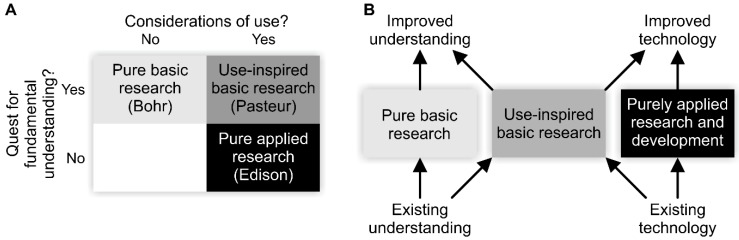
(**A**) The classification of research according to answers on two questions in Pasteur’s quadrant by Stokes [18]. The figure is redrawn from Figures 3–5 in [18], p. 73. The names of the eminent researchers Bohr, Pasteur, and Edison are meant to illustrate the different categories of research distinguished in this classification system. (**B**) The dynamic relationships between the different research approaches, redrawn from Figures 3–7 in [18], p. 88. Figures modified and reproduced with permission of the copyright owner.

**Table 1 animals-10-00213-t001:** A non-systematic inventory of publications on topics relating to feather pecking in laying hens. Research goals are classified using the type of research definitions in the Frascati Manual [17]. An expansion of this table is available as Appendix A and includes short descriptions of the studies’ experiments, subjects, measured parameters, results, and conclusions.

	Research Goals				
Objective	Pure Basic	Oriented Basic	Applied	Practical Application	Reference
**Environment**					
**Study the effects of BT and ground type during rearing on pecking behavior.**	No	Yes	Yes	Refine husbandry conditions to influence the expression of pecking behavior.	Blokhuis and van der Haar, 1989 [9]
**Investigate possible association between FP and stress.**	No	Yes	Yes	Refine husbandry conditions to control FP.	El-Lethey et al., 2000 [26]
**Investigate whether light exposure late in embryonic development affects early post-hatching FP behavior.**	No	No	Yes	Refine husbandry practices relating to light exposure during the last week of incubation.	Riedstra and Groothuis, 2004 [28]
**Investigate whether increasing dietary TRP decreases the development and performance of FP.**	No	No	Yes	Refine management strategies concerning diet choices to decrease the expression of FP.	van Hierden et al., 2004 [36]
**Study the effects of experience with an exploratory-rich environment during rearing on pecking behavior.**	No	No	Yes	Refine husbandry practices relating to enriched environments during rearing.	Chow and Hogan, 2005 [33]
**Investigate the effects of providing string devices on FP under (1) experimental and (2) commercial conditions.**	No	No	Yes	Refine husbandry conditions by offering device strings to reduce FP.	McAdie and Keeling 2005 [34]
**Investigate laying hen behavior under differential commercial stocking densities, flock sizes, and management practices.**	No	No	Yes	Refine management strategies to reduce the risk of FP.	Zimmerman et al., 2006 [30]
**Investigate the underlying motivation of GFP and SFP by comparing their motor patterns to those of dust bathing and foraging pecks.**	No	Yes	Yes	Refine husbandry conditions by offering forages (straw, hay, silage) to reduce SFP.	Dixon et al., 2008 [49]
**Investigate the role of environmental factors associated with the development of FP.**	No	No	Yes	Refine management practices to influence the expression of pecking behavior.	Lambton et al., 2010 [11]
**Study the effect of pen environment on group behavior and dynamics.**	No	Yes	Yes	Refine husbandry practices to better adjust to environmental needs of laying hens.	Collins et al., 2011 [25]
**(1) Investigate if dark brooders can successfully be used on commercial rearing farms. (2) Test if the subsequent FP reduction is replicable without compromising bird growth and mortality.**	No	No	Yes	Offer dark brooders as an alternative to standard husbandry practices to reduce FP.	Gilani et al., 2012 [32]
**(1) Investigate the effects of BT and EE during rearing on PD. (2) Study the relationship between behavior in the rearing period and PD in the laying period.**	No	No	Yes	Refine management practices to influence the expression of SFP.	Hartcher et al., 2015 [8]
**Study the effect of EE and a reduced stocking density on FP.**	No	No	Yes	Refine management practices to influence the occurrence of FP.	Zepp et al., 2018 [31]
**Genotype**					
**Examine differences between layer hen strains with regard to diurnal rhythm of FP and tendency to FP.**	No	No	Yes	Refine breeding and management strategies to reduce the risk of FP.	Kjear, 2000 [37]
**(1) Identify genotypes that show LFP in a free-range environment. (2) Study the effect of a diet enriched by sulfuric amino acids on FP. (3) Investigate the impact of light intensity during rearing in an interaction with access to the range area at different ages.**	No	No	Yes	Refine breeding and management strategies to reduce the risk of FP.	Kjaer and Sorensen, 2002 [29]
**Estimate heritability of FP and OF response of laying hens at different ages.**	No	No	Yes	Refine breeding strategies to reduce the risk of FP.	Rodenburg et al., 2003 [50]
**Study the effects of selection against mortality and BT on fear-related behavior and peripheral 5-HT concentration and uptake.**	No	No	Yes	Refine breeding and management strategies to reduce the risk of FP.	Bolhuis et al., 2009 [35]
**Determine parameters of heart rate variability in HFP and LFP lines to elucidate ANS responses during rest and stressful situations.**	No	Yes	Yes	Enable breeding efforts by mapping relationships between FP and ANS responses.	Kjaer and Jorgensen, 2011 [27]
**Investigate if sub-populations of EFP birds exist in HFP and LFP lines and their F2-cross.**	No	No	Yes	Refine breeding strategies to reduce the risk of FP.	Piepho et al., 2017 [38]
**Investigate if PS affects the development of anxiety and SFP in their offspring.**	No	No	Yes	Refine management practices to influence the expression of pecking behavior.	de Haas et al., 2014 [39]
**Examine relationships between the immune system and FP by characterizing HFP and LFP lines with regard to immune characteristics.**	No	Yes	Yes	Enable breeding efforts by exposing relationships between the immune system and FP.	van der Eijk et al., 2019 [40]
**Propose a new model to detect EFP: (1) Introduce a new theory and statistical method for the analysis of EFP; (2) define a new trait, EFPp; (3) analyze the interrelationship of EFPp with fearfulness.**	No	No	Yes	The identified new trait may refine breeding strategies to reduce EFP. An index of fear-related traits may serve as a proxy to breed indirectly for the new trait.	Iffland et al., 2019 [41]
**Phenotype**					
**Examine if tendency to avoid a NO was predictive of (1) pecking towards feather bundles or (2) FP. Analyze if pecking at feather bundles and FP were related.**	No	No	Yes	Offer a less time-consuming alternative to the current method of selecting LFP that is used for breeding.	Albentosa et al., 2003 [42]
**Investigate the relationship between fear responses, physiological measurements of basal plasma-CORT and whole-blood 5-HT, PD, and productivity in PS flocks.**	No	Yes	Yes	Refine management strategies concerning PS flocks by taking breed differences, group size effects, and effects of human–bird interactions into account.	de Haas et al., 2013 [43]
**Physiology**					
**Investigate if lowering 5-HT turnover in the forebrain of laying hens increases expression of FP.**	No	Yes	Yes	(1) Offer FP behavior as an animal model for an impulse-control disorder like trichotillomania. (2) Indicate that chronic enhancement of 5-HT neurotransmission in the chicken brain may be beneficial in reducing FP expression.	van Hierden et al., 2004 [44]
**Examine if immune modulation by airborne constituents predisposes birds for harmful behavior like FP.**	No	No	Yes	Refine vaccine management strategies to reduce the risk of FP.	Parmentier et al., 2009 [45]
**Study the neurobiological mechanisms of SFP.**	No	Yes	No	-	Kops et al., 2013 [46]
**Behavior**					
**Investigate the development of FP and related behaviors in chicks of HFP and LFP lines.**	No	Yes	No	-	Van Hierden et al., 2002 [47]
**Investigate if social learning is involved in the spread of cannibalism.**	No	Yes	No	-	Cloutier et al., 2002 [3]
**Investigate if coping theory can predict FP in laying hens.**	No	Yes	Yes	Discourage use of coping theory as a tool to find the underlying mechanisms of FP.	Forkman et al., 2004 [51]
**(1) Investigate HFP line preference for pecking at and/or ingestion of feathers over wood shavings and LPF line preference of wood shavings over feathers. (2) Investigate if hens housed alone in cages exhibit stronger motivation for their preferred substrate.**	No	Yes	Yes	Refine management strategies to influence the expression of FP.	Harlander-Matauschek et al., 2007 [48]

Abbreviations: serotonin (5-HT); autonomic nervous system (ANS); beak trimming (BT); corticosterone (CORT); environmental enrichment (EE); extreme feather pecking (EFP); feather pecking (FP); gentle feather pecking (GFP); high feather pecking (HFP); low feather pecking (LFP); novel object (NO); open field (OF); plumage damage (PD); parent-stock (PS); severe feather pecking (SFP); tryptophan (TRP).

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
