# Peer review of "Importance of Basic Research on the Causes of Feather Pecking in Relation to Welfare"

_animals, 2020, doi:10.3390/ani10020213_

Round 1

Reviewer 1 Report

This paper offers an interesting perspective on a very important and long-standing issue facing the commercial egg industry. The paper was well-written and easy to read. The authors provide well thought-out evidence for the argument and conclusions they are making.

MINOR COMMENTS

Line 29           change “focusses” to “focuses”

Line 118         change “pursuit” to “pursue”

Line 121         change “de” to “the”

Line 159         I think “the” is meant to be “they”?

Line 162         remove the second “that”

Line 169         I would change “unsystematic” to “non-systematic”

Line 256         Remove the “/”

Line 270         include abbreviation for gross domestic product (abbreviation is then used in Line 271)

MAJOR COMMENTS

Lines 62 – 79     The studies referenced in this paragraph either used primarily hot-blade trimming or were review papers on the effects of hot-blade trimming. Infrared beak treatment is a more recent and widely used method of beak treatment and is more welfare friendly compared to hot-blade trimming. This paragraph would benefit from the inclusion of studies examining the effect of infrared beak treatment.

McKeegan and Philbey. 2012 Animal Welfare 21: 207-217

Struthers et al. 2019 Poultry Science (2 papers)

Struthers et al. 2019 Animals 9: 665

Dennis et al. 2009 Poultry Science 88: 38-43

Dennis and Cheng 2010 International Journal of Poultry Science 9: 716-719

Dennis and Cheng 2012 Poultry Science 91: 1499-1505

Gentle and McKeegan 2007 Veterinary Record 160: 145-148

Marchant-Forde et al. 2008 Poultry Science 87: 1474-1483

Marchant-Forde and Cheng 2010 Poultry Science 89: 2559-2564

Line 73             When discussing the effects of beak trimming, the authors should state which method of beak trimming (hot-blade, infrared beak treatment) they are referring too. There is evidence that the negative effects of hot-blade trimming are not seen with infrared beak treatment. For example, the authors state that beak trimming causes chronic pain. However, in the review paper referenced, it mentions that infrared beak treatment is not associated with chronic pain. There are also papers that show that when hot-blade trimming is performed on young birds, there is no behavioural or histological evidence of neuroma formation or chronic pain

Schwean-Lardner et al., 2016 Journal of Applied Poultry Research 25: 547-560

Line 121           May be worthwhile to discuss what the current hypothesized causes of feather pecking are (e.g. redirected foraging, ground pecking, or dust bathing behaviours)

Line 145           Can you elaborate on what “chicken’s abilities” are?

Line 157           First time in manuscript “practice-based research” is mentioned. Is it the same as applied research? For clarity, it may be easier to use one term consistently throughout the manuscript.  

Author Response

Dear Reviewer 1,

Thank you for your consideration of our paper and your time to review it. I am grateful to your suggestions, as they helped to improve the manuscript. Attached to this reply you will find the revision with all changes highlighted in yellow. All but one of your suggestions were processed as a change in the text. Included in this message is a table where I will address your comments one by one.

Kind regards,

Lisa Fijn

Line (original)

Your comment

Line (revision)

Revision / reply

29           

change “focusses” to “focuses”

30

Changed to “focuses”

118        

change “pursuit” to “pursue”

124

Changed to “pursue”

121        

change “de” to “the”

128

Changed to “the”

159        

I think “the” is meant to be “they”?

170

Changed to “they”

162        

remove the second “that”

173

Second “that” removed

169        

I would change “unsystematic” to “non-systematic”

180

Changed to “non-systematic”

256        

Remove the “/”

266

Changed to “,”

270        

include abbreviation for gross domestic product (abbreviation is then used in Line 271)

282

Abbreviation for gross domestic product added

62 – 79    

The studies referenced in this paragraph either used primarily hot-blade trimming or were review papers on the effects of hot-blade trimming. Infrared beak treatment is a more recent and widely used method of beak treatment and is more welfare friendly compared to hot-blade trimming. This paragraph would benefit from the inclusion of studies examining the effect of infrared beak treatment.

73 – 82

Thank you for this suggestion and the literature that you provided. A reference is added to line 74 to specify that acute pain is associated with BT regardless of method used. In line 75 and 76 “by method of hot-blade” was added to specify that these effects follow the hot-blade method but have not been associated with the infrared treatment. Line 79-81 summarize some of the papers you suggested in saying that there has been a shift to the now widely used, less harmful, infrared treatment.

73            

When discussing the effects of beak trimming, the authors should state which method of beak trimming (hot-blade, infrared beak treatment) they are referring too. There is evidence that the negative effects of hot-blade trimming are not seen with infrared beak treatment. For example, the authors state that beak trimming causes chronic pain. However, in the review paper referenced, it mentions that infrared beak treatment is not associated with chronic pain. There are also papers that show that when hot-blade trimming is performed on young birds, there is no behavioural or histological evidence of neuroma formation or chronic pain

73-82

See previous reply.

121          

May be worthwhile to discuss what the current hypothesized causes of feather pecking are (e.g. redirected foraging, ground pecking, or dust bathing behaviours)

-

I agree that this is a worthwhile discussion and I did consider expanding on these hypothesizes for the original submission of the paper. I decided to leave this out however because the motivational factors for different types of FP vary and additionally there are many hypothesizes on the causes of FP and they sometimes contradict each other or differ from each other in more subtle ways than blatant contradictions. Expanding on the current hypothesizes and doing it properly would quickly result in a large body of text that would deflect from the arguments we are trying to present in this paper. Therefore, the choice was made to simply state that the exact causes in fact at this point remain largely unknown.

145          

Can you elaborate on what “chicken’s abilities” are?

154-155

Added “…chicken’s abilities, i.e. to the range of situations to which the animal is able to adapt to such an extent that it experiences positive welfare.”

157            

First time in manuscript “practice-based research” is mentioned. Is it the same as applied research? For clarity, it may be easier to use one term consistently throughout the manuscript.

168

Changed to “applied research”

Reviewer 2 Report

Dear authors

1. Based on your title "importance of basic research on causes of feather pecking in relation to welfare ", the results presented in your study seems not support your this point with none of those studies in literatures related to feather pecking is pure basic study but some of them are oriented basic studies. From this I could not recognize that this is the reason why is basic research so important to feather pecking? I can not see that logic.

2. From your results which are based on some research related to FP, but in my understanding the studies on FP over the past decades are enormous and many of them are related to mechanism or causes of FP, Are you sure your results covered all relevant studies in this field ?

Author Response

Dear Reviewer 2,

Thank you for your consideration of our paper and your time to review it. I would like to respond to your comments here.

In this opinion paper we argue for the importance of basic research on the causes of feather pecking relating to welfare. We argue that there is a need for more basic research. This is because we noticed that studies on feather pecking generally pertain applied research goals. In other words, studies on the topic of feather pecking in laying hens are mostly oriented towards finding solutions that can be applied in the setting of commercial farming, where incidences of feather pecking are highest. This general trend is illustrated by Table 1: at this moment there are more studies on feather pecking that pursue applied research goals and less studies that pursue basic research goals.While we feel that applied research on this matter is extremely valuable and should continue to receive much effort, we argue why basic research should receive corresponding effort, which it has not been receiving up to this point. You are right that there is an immense number of studies on feather pecking from the past few decades. From these there have been many insights into mechanisms and causes of feather pecking. Unfortunately, feather pecking still remains a continuing problem in laying hens, despite of all the research on the topic. We argue that an approach involving more basic research than it has involved up to this point, could help in establishing more well-rounded knowledge on the mechanisms and causes of feather pecking, and on how they relate to hen welfare. We have not done a systematic review of current literature. There are many prominent studies that are not displayed in Table 1. The table does not provide an all-embracing list but is meant as an illustration of the landscape of current literature on the topic of feather pecking by providing a sample of exemplary papers. We do not summarize all the literature on feather pecking, but we chose papers that we feel are representative of the current body of research on feather pecking.

Hopefully I have understood your questions and addressed them to your satisfaction here. Please let us know if you remain with any concerns or any unanswered questions.

Kind Regards,

Lisa Fijn

Reviewer 3 Report

This is an interesting paper and the authors do a well literature summary work. Just as you say, the basic research is as importmant as applied research for FP. Until now, the causes of  FP and cannibalism are still unclear. Your opinion provides the right direction for future research. But, can you provide a brief conclusion section to state the basic research direction you suppose , such as environmental or genetic factor and so on.

Author Response

Dear Reviewer 3,

Thank you for your consideration of our paper and your time to review it. Attached to this reply you will find the revision with all changes highlighted in yellow. The revision includes a conclusion (Line 298 – 313). In the conclusion we suggest that the basic research approaches should be directed towards questions concerning the adaptive capacity of chicken, their behavioral repertoire and needs, indices of negative and positive welfare states, neuronal and physiological mechanisms related to abnormal behavior, and factors such as gene by environment interactions, resulting in epigenetic modifications. We also state that research should focus on both the victim as well as the pecker, as both might experience compromised welfare. Further insight into the motivational systems underlying the multifactorial problem of abnormal behavior is needed in order to form a basis for prevention and intervention.

Kind Regards,

Lisa Fijn

Round 2

Reviewer 2 Report

Dear Authors

Thanks for your reply to my comments, and your explanation is very sound for me this time and I accept your opinion, and agreed it to be published in the present form.

Regards!